# Isolation and Identification of Dihydrophenanthrene Derivatives from *Dendrobium virgineum* with Protective Effects against Hydrogen-Peroxide-Induced Oxidative Stress of Human Retinal Pigment Epithelium ARPE-19 Cells

**DOI:** 10.3390/antiox12030624

**Published:** 2023-03-02

**Authors:** Pongsawat Panuthai, Rianthong Phumsuay, Chawanphat Muangnoi, Porames Maitreesophone, Virunh Kongkatitham, Wanwimon Mekboonsonglarp, Pornchai Rojsitthisak, Kittisak Likhitwitayawuid, Boonchoo Sritularak

**Affiliations:** 1Pharmaceutical Sciences and Technology Program, Faculty of Pharmaceutical Sciences, Chulalongkorn University, Bangkok 10330, Thailand; 2Department of Pharmacognosy and Pharmaceutical Botany, Faculty of Pharmaceutical Sciences, Chulalongkorn University, Bangkok 10330, Thailand; 3Center of Excellence in Natural Products for Ageing and Chronic Diseases, Chulalongkorn University, Bangkok 10330, Thailand; 4Cell and Animal Model Unit, Institute of Nutrition, Mahidol University, Nakhon Pathom 73170, Thailand; 5Scientific and Technological Research Equipment Centre, Chulalongkorn University, Bangkok 10330, Thailand; 6Department of Food and Pharmaceutical Chemistry, Faculty of Pharmaceutical Sciences, Chulalongkorn University, Bangkok 10330, Thailand

**Keywords:** *Dendrobium virgineum*, Orchidaceae, retinal pigment epithelium, oxidative stress, dihydrophenanthrene

## Abstract

Oxidative stress is a significant factor in the development of age-related macular degeneration (AMD), which results from cell damage, dysfunction, and death in the retinal pigmented epithelium (RPE). The use of natural compounds with antioxidant properties to protect RPE cells from oxidative stress has been explored in *Dendrobium*, a genus of orchid plants belonging to the family Orchidaceae. Two new compounds and seven known compounds from the MeOH extract of the whole plant of *Dendrobium virgineum* were successfully isolated and structurally characterized. Out of all the compounds isolated, 2-methoxy-9,10-dihydrophenanthrene-4,5-diol (**3**) showed the highest protective effect against hydrogen peroxide (H_2_O_2_)-induced oxidative stress in human retinal pigment epithelial (ARPE-19) cells. Therefore, it was selected to evaluate its protective effect and mechanism on oxidative-stress-induced ARPE-19 cells. Cells were pre-treated with compound 3 at 25, 50, and 100 µg/mL for 24 h and then induced with 400 µM H_2_O_2_ for 1 h. The results demonstrated that compound **3** significantly (*p* < 0.05) increased cell viability by 10–35%, decreased ROS production by 10–30%, and reduced phosphorylation of p38, ERK1/2, and SAPK/JNK by 20–70% in a dose-dependent manner without toxicity. Furthermore, compound **3** significantly (*p* < 0.05) modulated the expression of apoptosis pathway proteins (cytochrome c, Bax and Bcl-2) by 20–80%, and enhanced SOD, CAT, and GPX activities, and GSH levels in a dose-dependent manner. These results suggest that compound **3** protects ARPE-19 cells against oxidative stress through MAPKs and apoptosis pathways, including the antioxidant system. Thus, compound **3** could be considered as an antioxidant agent for preventing AMD development by protecting RPE cells from oxidative stress and maintaining the retina. These findings open up new possibilities for the use of natural compounds in the treatment of AMD and other oxidative-stress-related conditions.

## 1. Introduction

The abnormality of the eyes in the macular area causes irreversible blindness and retinal disease, the so-called age-related macular degeneration (AMD) [1]. The prevalence of AMD in Europe and Asia was around 200 million in 2020 and is expected to be a global health issue in 2040 [2,3,4,5]. AMD is primarily found in patients older than 60, with a loss of central vision that can disturb the patient’s daily life [6,7]. The pathogenesis of AMD is complex; however, it is divided into two types including dry or atrophic and wet or neovascular AMD [8]. The disease progression is associated with retinal pigment epithelium (RPE), which is necessary to maintain photoreceptor cells in the retina [7]. In elderly patients, RPE cannot maintain metabolic activity in the macula and clear the cellular waste and unusual proteins [7,9]. Eventually, it impairs the function of RPE, and overloaded waste is detected as drusen, a yellow spot in the retina, which is a vital sign of AMD [7,9,10]. Due to this imbalance, RPE loses function and affects the retina, which causes vision loss [11,12]. Various risk factors associated with AMD include genetic and exogenous factors such as aging, obesity, oxidative stress, and smoking [13,14]. According to the above risk factors, oxidative stress is the primary cause of the development of AMD [15].

Continuous exposure to blue light, which generates reactive oxygen species (ROS), is known to cause damage to RPE cells in the macula due to its ability to pass through the structures of the retina and induce photooxidation of lipofuscin to cytotoxic compounds [16]. RPE cells with abundant mitochondria exhibit elevated cellular metabolic rates in environments rich with endogenous reactive oxygen species, such as superoxide anions (O_2_−·) and hydroxyl radicals (OH·). ROS produced from mitochondria activity are also released into the cytoplasm and affect other cells, leading to an increase in ROS levels [17,18]. For the above reasons, high ROS production can be expected to lead to oxidative stress in RPE cells [19,20]. In various studies, ROS such as hydrogen peroxide (H_2_O_2_) cause the Fenton reaction, which is the chain reaction between iron in lysosomes and oxidants, resulting in more production of ROS, and are used to induce the oxidative stress condition in in vitro and in vivo [21,22].

Cellular dysfunction and abnormality of biomolecules, such as DNA damage and misfolding proteins, in cells are caused by the attack from ROS [23]. To survive, cells must remove the waste to maintain their function, especially in post-mitotic cells such as RPE cells [24]. The mitogen-activated protein kinases (MAPKs) signaling pathway, which controls the proliferation and cell death in RPE including Bax (pro-apoptotic) and Bcl-2 (anti-apoptotic), is activated by ROS [25,26]. Moreover, ROS can impact the mitochondrial outer membrane by altering the Bax and Bcl-2 proteins, which can lead to the release of cytochrome C into the cytoplasm, and activation of the apoptosis and caspase pathways, including caspase-3 and caspase-9 [27,28]. Such dysfunctions can cause damage to RPE cells and contribute to the development of AMD. Thus, the essential treatment for AMD focuses on the prevention of RPE cells from oxidative stress. Typically, RPE cells contain various endogenous antioxidants such as antioxidant enzymes including glutathione peroxidase (GPx), glutathione transferase (GST), catalase (CAT), superoxide dismutase (SOD), and the non-enzymatic antioxidant glutathione (GSH) for protection of the retina from oxidative damage [29]. A deficiency in antioxidant enzymes frequently accompanied by aging can contribute to the development of AMD [30].

Several studies on active compounds, such as kinsenoside from *Anoectochilus roxburghii* and herbal extracts such as *Emblica officinalis* (Amla) and bilberry anthocyanin extracts, have shown the existence of potent antioxidants for protective effects against AMD [31,32,33]. *Dendrobium* is a genus in the orchid family (Orchidaceae) with more than 1500 species worldwide [34]. These species have been proved to have therapeutic properties and have been used as folk medicines for more than 2300 years [35]. *Dendrobium* plants have been used as traditional medicine in China to treat fever, diabetes, stomach diseases, and lung and kidney disorders [36]. The natural exogenous non-enzymatic antioxidants such as polyphenols, carotenoids, flavonoids, and organosulfur compounds, which exhibited the preventive effects in AMD, have been reported in various studies [37,38,39,40,41,42,43,44,45]. Moreover, *Dendrobium* extract has been reported to improve vision [46]. *Dendrobium virgineum* Rchb.f. (Figure 1A), known in Thai as “Ueang Nang Chi”, is usually founded in dry evergreen forests and mixed deciduous forests in Thailand’s east and north-east regions, and in Myanmar, Laos, and Vietnam. Its flower consists of white petals and a red lip, and it was first discovered by Reichenbach in 1884 [47].

In this study, we aimed to isolate and determine the protective effect against H_2_O_2_-induced oxidative stress in human retinal pigment epithelial (ARPE-19) cells of isolated compounds from the whole plant of *Dendrobium virgineum*. Among **9** isolated compounds, two new compounds (**1** and **2**) and seven known compounds (**3**–**9**) were identified. Compound **3**, with the most potent antioxidant effect, was further investigated regarding its protective mechanism against H_2_O_2_-induced ARPE-19 cells.

## 2. Materials and Methods

### 2.1. Experimental

The Milton Roy Spectronic 3000 Array spectrophotometer (Rochester, Monroe, NY, USA) was used to record UV spectra, while IR spectra were obtained from the PerkinElmer FT-IR 1760X spectrophotometer (Boston, MA, USA). Mass spectra were determined using the Bruker MicroTOF mass spectrometer (ESI-MS) (Billerica, MA, USA). NMR spectra were recorded by the Bruker Avance DPX-300FT NMR spectrometer or the Bruker Avance III HD 500 NMR spectrometer (Billerica, MA, USA). Sigma Aldrich (Sigma-Aldrich, Dorset, UK) provided 2′,7′-dichlorofluorescein di-acetate (DCFH-DA), and Merck Millipore (Merck Millipore, Darmstadt, Germany) supplied fetal bovine serum (FBS). Invitrogen (Invitrogen Ltd., Paisley, UK) provided Dulbecco’s modified Eagle’s Medium/Nutrient Mixture F-12 Ham (DMEM/F-12), penicillin–streptomycin, H_2_O_2_, dimethyl sulfoxide (DMSO), and 3-[4,5-dimethyltiazol-2-yl]-2,5-diphenyl-tetrazolium bromide (MTT). Antibodies used in western blotting were obtained from Cell Signaling Technology (Danvers, MA, USA). Antioxidant enzyme assay test kits were obtained from Cayman Chemical (Cayman Chemical, Ann Arbor, MI, USA).

### 2.2. Plant Material

The whole plants of *D. virgineum* were obtained from Chatuchak Market (Bangkok, Thailand) in September 2015. Authentication was performed by the author (Boonchoo Sritularak). A voucher specimen (BS-DVir-2558) was deposited at the Department of Pharmacognosy and Pharmaceutical Botany, Faculty of Pharmaceutical Sciences, Chulalongkorn University.

### 2.3. Extraction and Isolation

The dried, powdered whole plant of *D. virgineum* (2.7 kg) was extracted in a stainless tank with 20 L of MeOH three times at room temperature to give a MeOH extract (297 g) after evaporation of the solvent. The MeOH extract was partitioned with 10 L of EtOAc and water to give an EtOAc extract (95 g) and an aqueous extract (200 g). The EtOAc extract was initially subjected to liquid chromatography under vacuum on silica gel (acetone–hexane, gradient) to yield 6 fractions (A-F). Fraction B (12.3 g) was separated by column chromatography (CC) over silica gel (acetone–hexane, gradient) to give 5 fractions (B−I to B−V). Fraction B−II (2.5 g) was fractionated by Sephadex LH-20 (MeOH) and then purified by CC (EtOAc–hexane, gradient) to yield 2-methoxy-9,10-dihydrophenanthrene-4,5-diol (**3**) (5 mg) and compound **1** (6 mg). Compound **1** (17 mg) and gigantol (**4**) (541 mg) were obtained from fractions B−III (553 mg) and B−IV (1.2 g), respectively, after purification on Sephadex LH-20 (MeOH). Fraction C (8.6 g) was separated by CC (silica gel, acetone–hexane, gradient) to give 9 fractions (C−I to C−IX). Fraction C−V (2.0 g) was fractionated by Sephadex LH-20 (MeOH) to give 4 fractions (C−V1 to C−V4). Fraction C−V2 (206 mg) was isolated by CC (silica gel, EtOAc–hexane, gradient) and further purified by Sephadex LH-20 (MeOH) to furnish compound **2** (6 mg). Fraction C−V3 (116 mg) was separated by CC (silica gel, EtOAc–hexane, gradient) and then subjected to repeated CC (silica gel, acetone–hexane, gradient) to yield 5-methoxy-7-hydroxy-9,10-dihydro-1,4-phenanthrenequinone (**5**) (4 mg). Fraction C−VI (1.3 g) was subjected to CC on silica gel (acetone–hexane) and then fractionated by Sephadex LH-20 (MeOH) to give 9 fractions (C−VI1 to C−VI9). Fraction C−VI2 (106 mg) was separated by CC (silica gel, EtOAc–hexane, gradient) to yield *p*-coumaric acid (**6**) (9 mg). Tristin (**7**) (101 mg) was obtained from fraction C−VI3 (237 mg) after purification on CC (silica gel, EtOAc–hexane, gradient). Purification of fraction C−VI5 (156 mg) by CC (silica gel, EtOAc–hexane, gradient) gave 2,5,7-trihydroxy-4-methoxy-9,10-dihydrophenanthrene (**8**) (116 mg). Fraction C−VI6 (11 mg) was separated by CC (silica gel, EtOAc–hexane, gradient) to give 2,4,7-trihydroxy-9,10-dihydrophenanthrene (**9**) (5 mg) (Figure 1).

Dendrovirginin (**1**): Brown amorphous solid; UV (MeOH) λmax (log ε): 222 (4.49), 273 (4.15), 305 (3.94) nm; IR (film) ν_max_: 3166, 2936, 2837, 1616, 1463, 1435, 1250, 1152, 1104 cm^−1^; HR-ESI-MS: [M−H]^−^ at *m/z* 271.0960 (calcd. for 271.0970, C_16_H_15_O_4_); ^1^H NMR (500 MHz, acetone-*d*_6_) and ^13^C NMR (125 MHz): see Table 1.

Dendrovirginone (**2**): Red amorphous solid; UV (MeOH) λmax (log ε): 222 (4.39), 260 (4.07), 334 (3.79), 490 (3.53) nm; IR (film) ν_max_: 3416, 2939, 2922, 1722, 1658, 1605, 1560, 1349, 1213, 1160 cm^−1^; HR-ESI-MS: [M−H]^−^ at *m/z* 285.0752 (calcd. for 285.0763, C_16_H_13_O_5_); ^1^H NMR (500 MHz, acetone-*d*_6_) and ^13^C NMR (125 MHz): see Table 1.

### 2.4. Culture of ARPE-19 Cells

ARPE-19 cells (ATCC, Rockville, MD, USA) were used as a representative model of human RPE cells and cultured in DMEM/F-12 with 10% (*v*/*v*) heat-inactivated fetal bovine serum (FBS) and 1% (*v*/*v*) penicillin–streptomycin (pen–strep). Standard 6-well or 96-well plates were used for all experiments, unless otherwise stated. Cells were seeded at 1.0 × 10^6^ and 3.0 × 10^4^ cells/well for 6- and 96-well plate experiments, respectively. The cells were allowed to grow until they reached over 90% confluence before being utilized in subsequent experiments.

### 2.5. Cell Viability Assay

The viability of cells was assessed using the 3-(4,5-dimethylthiazol-2-yl)-2,5-diphenyltetrazolium bromide tetrazolium (MTT) assay, which measures the conversion of MTT substrate to a purple-colored formazan product in viable cells. After cells reached their experimental time points, they were washed once with phosphate buffer saline (PBS) and incubated with MTT in PBS at 0.5 mg/mL for 4 h at 37 °C. After removal of the MTT solution, DMSO was added to dissolve the formazan crystals generated by viable cells. The absorbance was subsequently measured at 540 nm using a microplate reader (SPECTROstarNano, BMG LABTECH, Ortenberg, Germany).

### 2.6. Cytotoxicity Assay of Compounds ***1***–***9***

Following a 24 h incubation of ARPE-19 cells, the media was removed and cells were washed with serum-free media. Cells were then treated with varying concentrations (10, 50, and 100 µg/mL) of compounds **1**–**9** for 24 h, with a 0.5% DMSO used as the control group. Cell viability was assessed using the MTT assay after the respective experimental time.

### 2.7. Determining the Optimal H_2_O_2_ Concentration for Cytotoxicity Induction

ARPE-19 cells in a 96-well plate were treated by removing the media and washing with serum-free media. Subsequently, cells were treated with serum-free medium containing various concentrations of H_2_O_2_ (200–1000 µM) for 30, 60, and 120 min at 37 °C. The control group consisted of cells cultured in serum-free medium without H_2_O_2_. Following the experimental treatment, cells were washed twice with PBS and evaluated for cell viability using the MTT assay.

### 2.8. Assessing the Effect of Compound ***3*** on ARPE-19 Cells Exposed to H_2_O_2_

ARPE-19 cells in 6-well and 96-well plates were washed one time with serum-free media and pre-incubated with compound **3** (25, 50, and 100 µg/mL) in serum-free media for 24 h. A 0.5% DMSO served as the control group. After pre-treatment with compound **3**, cells were washed with serum-free media and then treated with appropriate H_2_O_2_ concentrations in serum-free medium at 37 °C for 1 h. Cell viability was measured using the MTT assay in the 96-well plate setup, while the 6-well plate setup was used for non-enzymatic and enzymatic antioxidant assays, caspase-9 and caspase-3 activities, and western immunoblot analysis.

### 2.9. Evaluation of Reactive Oxygen Species (ROS) Production

To determine intracellular ROS, ARPE-19 cells were seeded at 3.0 × 10^4^ cells/well in black 96-well, clear bottom plates and cultured for 24 h. After washing with serum-free media, cells were pre-incubated for 24 h with compound **3** (25, 50 and 100 µg/mL) in serum-free media, with 0.5% DMSO as the control. Subsequently, cells were incubated with 10 µM DCFH-DA in serum-free media at 37 °C for 30 min, then treated with varying concentrations of H_2_O_2_ in serum-free media at 37 °C for 1 h. Following PBS washes, ROS production was measured using a fluorescence microplate reader with excitation/emission wavelengths of 485/530 nm.

### 2.10. Western Blot Analysis

ARPE-19 cell lysates were prepared from a 6-well plate by centrifuging at 14,000 rpm at 4 °C for 10 min, and the protein concentrations were determined using the BCA protein assay kits. Equal amounts (40 µg) of protein samples were separated by 10% SDS-PAGE, transferred to a nitrocellulose membrane, blocked with 5% dry milk, and then incubated with anti-p-p38, anti-p-ERK1/2, anti-p-SAPK/JNK, anti-Bax, anti-Bcl-2, or anti-cytochrome c at a ratio of 1:1000 (*v*/*v*) in TBST overnight. The membrane was washed with TBST and incubated with a 1:2000 species-specific horseradish peroxide conjugated secondary antibody for 2 h. The protein levels were detected using an enhanced chemiluminescent detection kit, followed by imaging. The densitometry values of the phosphorylated forms of p38 (p-p38), ERK1/2 (p-ERK1/2), and SAPK/JNK (p-SAPK/JNK) were normalized to the band intensity of the respective total forms of p38, ERK1/2, and SAPK/JNK. Similarly, the densitometry values of cytochrome c, Bax, and Bcl-2 were normalized to the band intensity of β-actin.

### 2.11. Caspase-9 and -3 Activities

Following pre-treatment with compound **3** and H_2_O_2_ induction, as described in Section 2.8, the cells were homogenized in a hypotonic buffer to extract the supernatant. The supernatant was combined with a specific substrate (N-acetyl-Leu-Glu-His-Asp p-nitroanilide or N-acetyl-Asp-Glu-Val-Asp p-nitroanilide for caspase-9 or caspase-3, respectively) at a concentration of 100 µmol/L. The mixture was incubated at 37 °C for 1 h., and the absorbance was measured at 450 nm using a microplate reader to detect the activity of the caspases.

### 2.12. SOD, GPx, CAT, and GSH Determination

After treating ARPE-19 cells with compound **3** and H_2_O_2_, as described in Section 2.8, the cells were harvested by scraping and incubated with 0.5% (*v*/*v*) Triton X-100 in cold PBS. The resulting cell solution was transferred to a 1.5-mL tube and subjected to sonication in an ultrasonic sonicator bath at 4 °C for 10 min. The cell lysate was then centrifuged at 14,000× *g* at 4 °C for 10 min, and the supernatant was collected to measure the levels of GSH and the activities of SOD, GPx, and CAT using assay kits.

### 2.13. Statistical Analysis

The results are expressed as mean ± standard deviation (SD) of at least three independent experiments. Statistical analysis was performed with SPSS software version 16.0 (SPSS inc., Chicago, IL, USA). The differences among groups were assessed by one-way analysis of variance (ANOVA). Statistical significance was set at *p* < 0.05.

## 3. Results

### 3.1. Structural Characterization

Chromatographic separation of EtOAc extract from the whole plants of *D. virgineum* resulted in the isolation of two previously unknown phenanthrene derivatives (**1** and **2**), along with seven known compounds, which included 2-methoxy-9,10-dihydrophenanthrene-4,5-diol (**3**) [48], gigantol (**4**) [49], 5-methoxy-7-hydroxy-9,10-dihydro-1,4-phenanthrenequinone (**5**) [50], *p*-coumaric acid (**6**) [51], tristin (**7**) [52], 2,5,7-trihydroxy-4-methoxy-9,10-dihydrophenanthrene (**8**) [53], and 2,4,7-trihydroxy-9,10-dihydrophenanthrene (**9**) [54] (Figure 1B).

Compound **1** was purified as a brown amorphous solid. A molecular formula C_16_H_16_O_4_ was deduced from its [M−H]^−^ at *m/z* 271.0960 (calcd for C_16_H_15_O_4_ 271.0970). The IR spectrum exhibited absorption bands for hydroxyl (3166 cm^−1^), aromatic (2936, 1616 cm^−1^), and methylene (1463 cm^−1^) functionalities. The UV absorptions at 222, 273, and 305 nm indicated the characteristic of a dihydrophenanthrene nucleus [55]. This was confirmed by the presence of four methylene protons at δ 2.58–2.65 (4H, m, H_2_-9, H_2_-10), which showed correlations to the carbon atom at δ 23.0 (C-9) and δ 31.6 (C-10). The ^1^H NMR displayed four aromatic protons at δ 6.45–6.88 and two methoxy groups at δ 3.77 (3H, s, MeO-2) and 3.78 (3H, s, MeO-8). On ring A, the ^1^H NMR showed two doublet proton signals at δ 6.45 (1H, d, *J* = 2.5 Hz, H-3) and 6.49 (1H, d, *J* = 2.5 Hz, H-1). The assignment of H-1 was based on its HMBC correlation with C-10 (δ 31.6) and NOESY interaction with H_2_-10 (Figure 2). The first methoxy group was located at C-2 according to its NOESY correlations with H-1 and H-3. A comparison of ^1^H and ^13^C NMR of ring B of **1** with those of dendroinfundin B, a dihydrophenanthrene derivative previously reported from *Dendrobium infundibulum* [56], revealed their structural similarity by the presence of two doublet protons at δ 6.84 (1H, d, *J* = 8.5 Hz, H-7) and 6.88 (1H, d, *J* = 8.5 Hz, H-6), and a methoxy group at δ 3.78 (3H, s, MeO-8). The assignment of H-7 was according to 3-bond correlations of C-8a (δ 129.3) with H-7 and H_2_-10. The second methoxy group was substituted at C-8 based on its NOESY correlation with H-7 and H_2_-9 (Figure 2). Based on the above spectral evidence, **1** was characterized as 4,5-dihydroxy-2,8-dimethoxy-9,10-dihydrophenanthrene and named dendrovirginin.

Compound **2**, a red amorphous solid, was analyzed for C_16_H_14_O_5_ from its [M−H]^−^ at *m/z* 285.0752 (calcd for C_16_H_13_O_5_ 285.0763). The IR spectrum showed absorption bands for hydroxyl (3416 cm^−1^), aromatic (2939, 1658 cm^−1^), and ketone (1722 cm^−1^) functionalities. The UV absorptions at 222, 260, 334, and 490 nm suggest a dihydrophenanthrenequinone nucleus [50]. This was supported by the presence of the ^1^H NMR signals for two pairs of methylene protons at δ 2.46 (2H, br s, H_2_-10) and 2.58 (2H, t, *J* = 7.0 Hz, H_2_-9), and the ^13^C NMR signals of carbonyl carbon at δ 180.7 (C-1) and 185.1 (C-4). Compound **2** showed ^1^H and ^13^C NMR resonances similar to those of 5-methoxy-7-hydroxy-9,10-dihydro-1,4-phenanthrenequinone (**5**), a dihydrophenanthrenequinone also isolated from this plant, except for the presence of a methoxy group (δ 3.82, 3H, s) at C-2 of 2. The substitution of this methoxy group was supported by the presence of a sharp singlet proton signal of H-3 (δ 5.95, 1H, s), which showed correlations with C-1 (δ 180.7) and C-4a (δ 142.1), and the NOESY interaction between MeO-2 and H-3 (Figure 2). The ^1^H NMR also showed two doublet protons at δ 6.41 (1H, d, *J* = 2.0 Hz, H-8) and 6.43 (1H, d, *J* = 2.0 Hz, H-6), and a methoxy group at δ 3.69 (3H, s, MeO-5). The HMBC correlation of H-8 with C-9 and the NOESY correlation of H-8 with H_2_-9 were also observed (Figure 2). The placement of MeO-5 was supported by its NOESY correlation with H-6. Therefore, compound **2** was identified as 7-hydroxy-2,5-dimethoxy-9,10-dihydro-1,4-phenanthrenequinone and given the trivial name dendrovirginone.

### 3.2. Evaluation of the Effects of Compounds (***1***–***9***) on Viability of ARPE-19 Cells

To measure their non-toxic concentrations, compounds **1**–**9** were tested on ARPE-19 cells before assessing their protection against oxidative stress. The treatment was conducted for 24 h at 25, 50, and 100 µg/mL. Compounds **1**, **2**, and **4** at 50 and 100 µg/mL, as well as compounds **8** and **9** at 100 µg/mL, exhibited cytotoxicity against ARPE-19 cells (Figure 3). To ensure efficient and continuous activity, the maximum concentration used in subsequent experiments was 25 µg/mL.

### 3.3. Evaluation of the Effect of H_2_O_2_ on Viability and ROS Production of ARPE-19 Cells

Various concentrations of H_2_O_2_ (200–1000 µM) were applied to ARPE-19 cells for 30, 60, and 120 min to determine the concentration required for a roughly 50% reduction in viability. Results indicated that H_2_O_2_ treatment caused a concentration and time-dependent decrease in cell viability and an increase in ROS production (Figure 4A,B). Treatment with 400 µM of H_2_O_2_ for 60 min caused a 50% reduction in cell viability (Figure 4A). As a result, 400 µM of H_2_O_2_ for 60 min was employed to generate oxidative stress in ARPE-19 cells.

### 3.4. Evaluation of the Effect of Compounds (***1***–***9***) on Cell Viability of Oxidative-Stress-Induced ARPE-19 Cells

To assess their protective effects against H_2_O_2_-induced oxidative stress in ARPE-19 cells, compounds **1**–**9** were evaluated by pre-incubating cells with each compound at 25 µg/mL for 24 h. After washing with serum-free media, cells were treated with serum-free media containing 400 µM of H_2_O_2_ for 60 min. Among the compounds, compound **3** showed the highest protective effect against oxidative stress in ARPE-19 cells (Figure 5A) without inducing toxicity in normal ARPE-19 cells (Figure 5B). Therefore, compound **3** was selected to evaluate its protective mechanism in oxidative-stress-induced ARPE-19 cells. As shown in Figure 3, the cytotoxicity results revealed that the maximum concentration of compound **3** at 100 µg/mL had no significant impact on viability of ARPE-19 cells compared with the control. Consequently, concentrations of 25, 50, and 100 µg/mL of compound **3** were chosen for the protective mechanism studies.

### 3.5. Evaluation of the Effect of Compound ***3*** on Cell Viability and ROS Production in Oxidative-Stress-Induced ARPE-19 Cells

The protective effects of compound **3** against oxidative-stress-induced cell death in ARPE-19 cells were investigated by pre-incubation with compound **3** at 25, 50, and 100 µg/mL for 24 h followed by induction of oxidative stress with 400 µM H_2_O_2_ for 60 min. The protective effect of compound **3** against H_2_O_2_-induced oxidative stress was supported by inverted microscopic analysis (Figure 6A). H_2_O_2_ treatment caused a 50% decrease in cell viability compared with the control group (Figure 6B). However, compound **3** significantly (*p* < 0.05) protected the cell viability of ARPE-19 cells in a dose-dependent manner when compared with the H_2_O_2_ group (Figure 6B). In terms of ROS production, H_2_O_2_ significantly (*p* < 0.05) increased ROS production compared with the control group (Figure 6C). On the other hand, compound **3** significantly (*p* < 0.05) decreased ROS production in a dose-dependent manner when compared with the H_2_O_2_ group (Figure 6C). These findings suggest that compound **3** protects ARPE-19 cells against oxidative-stress-induced cytotoxicity by reducing ROS production dose-dependently.

### 3.6. Evaluation of the Effect of Compound ***3*** on MAPKs Protein Expression in Oxidative-Stress-Induced ARPE-19 Cells

The protective effects of compound **3** on ARPE-19 cells under oxidative stress were investigated to determine the underlying molecular mechanisms. Previous research indicated that phosphorylation of MAPK signaling pathways (p38, ERK1/2, and SAPK/JNK) promoted H_2_O_2_-induced apoptosis [57]. The current study explored whether the same pathway contributed to H_2_O_2_-induced cell damage and death. Immunoblotting was used to analyze protein expression and revealed that incubation of ARPE-19 cells with H_2_O_2_ at 400 µM for 1 h significantly increased the phosphorylation of p38 (p-p38), ERK1/2 (p-ERK1/2), and SAPK/JNK (p-SAPK/JNK) in comparison with the control group (Figure 7A–C). This finding suggests that H_2_O_2_-induced cell death occurs via the p-p38, p-ERK1/2, and p-SAPK/JNK pathways. Pre-incubation with compound **3** (25, 50, and 100 µg/mL) for 24 h reduced the expression of the phosphorylation form of p38, ERK1/2, and SAPK/JNK in comparison with the H_2_O_2_-induced ARPE-19 cell group (*p* < 0.05), indicating that compound **3** protects against oxidative stress via a dose-dependent modulation of the MAPKs signaling pathway.

### 3.7. Evaluation of the Effect of Compound ***3*** on Apoptosis Protein Expression in Oxidative-Stress-Induced ARPE-19 Cells

To understand how compound **3** protects against oxidative stress, we assessed its molecular mechanisms by analyzing the apoptosis pathway. Specifically, we evaluated the levels of downstream targets of the MAPK pathways, such as cytochrome c, Bax (pro-apoptotic), and Bcl-2 (anti-apoptotic) proteins [58,59]. We determined the expression of cytochrome c, Bax, and Bcl-2 through immunoblotting in ARPE-19 cells induced with oxidative stress (Figure 8A–C). Our results demonstrated that H_2_O_2_ incubation significantly increased cytochrome c and Bax levels and decreased Bcl-2 levels compared with the control group.

However, pre-incubating cells with compound **3** at concentrations of 25, 50, and 100 µg/mL for 24 h resulted in a significant dose-dependent decrease in the levels of cytochrome c and Bax proteins, and a significant dose-dependent increase in the level of Bcl-2 protein compared with the H_2_O_2_ group (Figure 8A–C). Thus, in a dose-dependent manner, the protective effect of compound **3** against oxidative-stress-induced cell death is mediated by regulating the cytochrome c, Bax, and Bcl-2 proteins in the apoptosis pathway.

### 3.8. Effect of Compound ***3*** on Caspase-9 and Caspase-3 Activities in ARPE-19 Cells under Oxidative Stress

To investigate the anti-apoptotic effect of compound **3** on oxidative-stress-induced ARPE-19 cells, we examined its impact on caspase-9 and caspase-3 activities. H_2_O_2_ exposure significantly (*p* < 0.05) increased the activity of both caspases compared with the control group (Figure 9A,B). However, pre-incubation with compound **3** at 25, 50, and 100 µg/mL for 24 h significantly decreased caspase-9 and caspase-3 activities in a dose-dependent manner compared with the H_2_O_2_ group (Figure 9A,B). These findings indicate that compound **3** can mitigate H_2_O_2_-induced apoptosis from oxidative stress by modulating the apoptosis pathway via caspase-9 and caspase-3 activities.

### 3.9. Evaluation of the Effect of Compound ***3*** on SOD, CAT, and GPx Activities as well as GSH Levels in ARPE-19 Cells under Oxidative Stress

To determine whether compound **3** modulates enzymatic antioxidants such as superoxide dismutase (SOD), catalase (CAT), and glutathione peroxidase (GPx), as well as the non-enzymatic antioxidant glutathione (GSH), we examined their activities and levels in ARPE-19 cells under oxidative stress. Results showed that H_2_O_2_ incubation significantly decreased SOD, CAT, and GPx activities, and GSH levels compared with the control group (*p* < 0.05) (Figure 10A–D). However, pre-incubation of cells with compound **3** at concentrations of 25, 50, and 100 µg/mL for 24 h significantly improved SOD, CAT, and GPx activities, and GSH levels in a dose-dependent manner in ARPE-19 cells induced with oxidative stress (Figure 10A–D). Interestingly, incubation with only compound **3** at a concentration of 100 µg/mL significantly increased SOD, CAT, and GPx activities, and GSH levels compared with the control group (Figure 10A–D). These findings suggest that pre-treatment with compound **3** could enhance the antioxidant system in RPE cells and protect against potential oxidative stress inducers. The findings suggest that prior administration of compound **3** could enhance the antioxidant system of ARPE-19, thus providing protection against potential triggers of oxidative stress.

## 4. Discussion

Plants in the genus *Dendrobium* have been traditionally used for medicinal purposes, and one of the benefits of *Dendrobium* extract is vision improvement [46]. In this study, we initially investigated the protective effects of the MeOH extract from the whole plants of *D. virgineum* on H_2_O_2_-induced oxidative stress in ARPE-19 cells. Chromatographic isolation of this plant led to the isolation of two new compounds, dendrovirginin (compound **1**) and dendrovirginone (compound **2**), along with seven known compounds (compounds **3–9**). The isolated compounds from the plant were subsequently evaluated for their cytotoxicity and protective effects on ARPE-19 cells under oxidative stress. Among the isolated compounds, only compound **3** showed the highest protective effect without inducing toxicity. Therefore, compound **3** could be a potential candidate for treating oxidative-stress-related eye diseases and was chosen for further protective effect and mechanism evaluation.

This study represents the first demonstration of the protective effects of the natural bioactive compound, compound **3**, belonging to the dihydrophenanthrene group, isolated from *D. virgineum*, against oxidative stress in ARPE-19 cells. Compound **3** exhibited its protective effect by modulating the key apoptotic mitogen-activated protein kinases (MAPKs), namely p38, extracellular-signal-regulated kinases 1/2 (ERK1/2 or p44/42), and stress-activated protein kinases/c-Jun N-terminal kinases (SAPK/JNK), as well as the apoptotic signaling pathway, encompassing Bax, Bcl-2, and cytochrome c. Furthermore, compound **3** could protect ARPE-19 under oxidative stress by enhancing the activities of enzymatic antioxidant systems, including SOD, CAT, and GPx, and the non-enzymatic antioxidant GSH. Previous studies have demonstrated that natural bioactive compounds belonging to the dihydrophenanthrene group possess direct antioxidant activities in 2,2-diphenyl-1-picrylhydrazyl (DPPH), 2,2′-azino-bis(3-ethylbenzothiazoline-6-sulfonic acid) (ABTS), and cupric ion reducing antioxidant capacity (CUPRAC) assays [60]. Hence, it is plausible that compound **3** may confer its protective effect on oxidative stress in ARPE-19 cells via direct antioxidant activity. Taken together, these findings emphasize the potential utility of compound **3** as a prophylactic agent for AMD.

Moreover, we found that compound **3** exerts a protective effect via the modulation of phosphorylation of MAPKs signaling, namely p38, ERK1/2, and SAPK/JNK. These MAPKs play essential roles in cellular functions such as apoptosis and proliferation [58,59]. Transient or acute stimulation of this pathway is crucial for normal cell survival, whereas sustained or chronic stimulation can lead to cell death. Such studies show that stimulating these MAPKs can cause downstream expression of apoptotic regulators, including Bax (pro-apoptotic), Bcl-2 (anti-apoptotic), and cytochrome c. In addition, the phosphorylation of MAPKs is strongly related to the promotion of H_2_O_2_-induced cell apoptosis and death in RPE cells [59,61,62]. Hence, assessing the activation of MAPK and the expression of downstream molecules such as Bax, Bcl-2, and cytochrome c provides insights into the mechanism of H_2_O_2_-induced apoptosis in ARPE-19. The treatment of H_2_O_2_ led to an increase in the phosphorylation of p38, ERK1/2, and SAPK/JNK in ARPE-19. As a result, there was an increase in Bax and cytochrome c expression, along with a decrease in Bcl-2 expression, indicating that cell death following H_2_O_2_ treatment occurred via the apoptotic signaling pathway. Pre-incubation with compound **3** prevented these changes, leading to increased cell viability under oxidative stress by H_2_O_2_. These findings highlight the potential of compound **3** as a preventive agent for oxidative-induced cell dysfunction and death in RPE cells, particularly in the case of AMD.

Additionally, pre-incubation with compound **3** increased the activities of key enzymatic antioxidants such as superoxide dismutase (SOD), catalase (CAT), and glutathione peroxidase (GPx), as well as the levels of the non-enzymatic antioxidant glutathione (GSH). These results suggest that compound **3** can function indirectly as an antioxidant by enhancing the activities and levels of these critical antioxidants. Under oxidative stress conditions induced by incubation with H_2_O_2_, we observed a decrease in the activities of SOD, CAT, and GPx, as well as the levels of GSH. However, pre-incubation with compound **3** improved the activities of these enzymes and the level of GSH in comparison with the cells under oxidative stress. Furthermore, under normal conditions, where the cells were pre-incubated with compound **3** without H_2_O_2_, we observed a significant increase in SOD, CAT, and GPx activities, as well as GSH levels compared with the control group. These results indicate that the pre-treatment of compound **3** could support oxidative stress protection by enhancing the SOD, CAT, and GPx activities, and the levels of GSH. A previous study showed that bioactive compounds from the *Dendrobium* extract protected HaCaT keratinocytes cells from oxidative stress by activating non-enzymatic and enzymatic antioxidant systems, leading to reduced ROS production [63]. Another study showed that the bioactive compounds in the dihydrophenanthrene group increased enzymatic antioxidant activity in polymorphonuclear leukocytes [64].

The exact mechanisms by which compound **3** influences SOD, CAT, and GPx activities, as well as the GSH level, were not evaluated in this study. However, previous studies have reported that antioxidant compounds can protect RPE cells against oxidative stress by activating the signaling pathway of Akt/Nrf2 [65,66]. This pathway relates the translocation of Nrf2 into the nucleus, which leads to the expression of various non-enzymatic and enzymatic antioxidants. Numerous studies have reported that dihydrophenanthrene compounds, such as compound **3**, exert protective effects against oxidative stress by activating the Nrf2 signaling pathway [67,68]. Therefore, it is likely that compound **3** can influence the levels of the non-enzymatic and enzymatic antioxidants through this pathway. Our findings suggest that compound **3** can protect RPE cells from oxidative stress by enhancing the activities of key enzymatic antioxidants, such as SOD, CAT, and GPx, and the level of GSH. Further studies are needed to elucidate the precise mechanisms by which compound **3** influences these antioxidants’ activities and levels.

## 5. Conclusions

In conclusion, the study shows that compound **3**, a dihydrophenanthrene group compound isolated from *Dendrobium virgineum*, has a protective effect against oxidative-stress-induced damage in ARPE-19 cells by modulating key apoptotic signaling pathways and enhancing the activities of enzymatic and non-enzymatic antioxidants (Figure 11). These findings suggest that compound **3** has potential as a prophylactic agent for AMD and other oxidative-stress-related eye diseases. The study also highlights the importance of evaluating the activation of MAPKs and the apoptosis pathway to understand how oxidative stress induces apoptosis in RPE cells. The discovery of new natural bioactive compounds such as compound **3** provides opportunities for the development of novel therapeutic agents to prevent and treat oxidative-stress-induced diseases.

## Data Availability

The data are contained within the article.

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
