# Peer review of "Isolation and Identification of Dihydrophenanthrene Derivatives from Dendrobium virgineum with Protective Effects against Hydrogen-Peroxide-Induced Oxidative Stress of Human Retinal Pigment Epithelium ARPE-19 Cells"

_antioxidants, 2023, doi:10.3390/antiox12030624_

Round 1

Reviewer 1 Report

Dear authors,

Let me start by congratulating you on the article. I found the study to be interesting, but I think there are a few issues that should be corrected before this manuscript is published.

Title: There is a discrepancy between the title, the objectives of the study presented in the introduction and the structure of the experiment. I finally realized after reading the manuscript that there were just two objectives: identification of the components from the alcohol extract obtained from the entire plant, and assessment of the biological effects for one of the nine identified compounds. I think there needs to be consistency between the title, objectives, and outcomes. This connection must be reflected in the manuscript's title.

Abstract: I believe that the chapter is clearly presented but some corrections are required. In my opinion, the abstract should contain statistically significant data obtained in the determinations.

Introduction

Some corrections and additional data are required.

Line 66 – specify what kind of light.  Blue component of visible light, which passes through the structures of the retina, is the most harmful to the RPE (because it causes the photooxidation of lipofuscin to cytotoxic compounds).

Lines 66 – 75 – The topic is correctly presented by some citations, which are relatively old for this dynamic field such as oxidative stress and RPE cells. I believe that the cited literature needs to be updated.

Line 81: « ROS also infest …. » I don't think it's appropriate. Please rephrase.

Line 90: From the category of exogenous non-enzymatic antioxidants, several review articles have been published in recent years, which showed the importance of polyphenols and carotenoids in the prevention of Age-related macular degeneration. Perhaps it should be mentioned, in the introduction, the importance of these compounds so widespread in the human diet. Some examples of such articles: https://www.hindawi.com/journals/omcl/2019/9682318/ si https://www.hindawi.com/journals/omcl/2019/9783429/ (published in 2019) ; https://www.mdpi.com/2072-6643/14/4/827 (published in 2022).

Materials and Methods

Lines 135 – 161: Keeping track of and visualizing the many phases and fractions is really challenging. I think that a well-organized scheme (image) would make it easier for the readers to understand and follow this complex protocol.

Results and Discussions

From the data presented previously, 2 new compounds were identified (compound 1 and 2), the rest being compounds that were identified in this plant and previously tested. Why did the analysis focus on compound 3 alone? I don't understand. Why were the newly discovered compounds (1 and 2) excluded from the investigation?

Figure 6A – the images are very small, blurry.

Figures 6B and 6C: First column Cont = Control group? What does the last column mean? Please select a more understandable way to display the data in these graphs.

Figure 7: The images of the bands are blurry. Please insert images with better resolution. For graphs, similar observations to those in figures 6A and 6B. Figures 8, 9 and 10: see previous comments.

Reviewer 2 Report

The manuscript by Panuthai et al. describes the isolation of biologically active substances, including dihydrophenanthrene derivatives, from Dendrobium virgineum. Two previously unknown compounds were obtained and characterized, along with seven known compounds. It was shown that one of the substances (Compound 3) is efficient against oxidative stress-induced damage in ARPE-19 cells. The results are important and demonstrate the prospects of developing new therapeutics for treatment of age-related macular degeneration and other oxidative stress-induced diseases.

Specific comments:

Abstract:  A full name of compound should be mentioned at least one time in Abstract along with the compound number (3) .

Line 33:  Perhaps, the phrase "treatment of compound 3" should be replaced with "treatment with compound 3". Please check the sense of the sentence.

Line 69:  Please use correct subscripts and superscripts in the designations of superoxide anion and hydroxyl radical.  In the current version of manuscript, a monoatomic oxygen dianion and hydroxide anion are shown instead.

Section 2.3 (Extraction and Isolation):  The extraction should be described in more details. Please indicate total volumes of extractants and apparatus used. 

Line 147:  Using Roman numbers along with letter "C" is confusing. For example, the fraction designation "CVI" can be misunderstood as the Roman number "106", etc.

Table 1:  The unit (Hz) for NMR signal multiplicity is given in the column headers. It is not necessary to repeat this unit through the table.

Line 259:  Please replace "unknown" with "previously unknown".

Line 269:  The statement "indicated a dihydrophenanthrene nucleus" should be replaced with a more cautious phrase like "is characteristic of a dihydrophenanthrene nucleus" or similar.

Line 498:  It is reasonable to replace "underscore" with "emphasize". 

Summarizing, I recommend acceptance of the manuscript for publication after minor revision.

Round 2

Reviewer 1 Report

Dear authors,

I read the manuscript and noticed that you made the requested changes. The quality of the article resubmitted has increased.